# Non-Myopic Batch Bayesian Experimental Design for Quantifying Statistical Expectation

## 1   Introduction

For almost a century, the fundamental method to estimate statistical expectation has been Monte Carlo with the core idea of learning a system by many random samples [1]. Although the convergence of Monte Carlo is guaranteed by the *law of large numbers*, its convergence rate—inversely proportional to the square root of the number of samples—is notoriously slow. That becomes a problem in scenarios where systems, such as global weather or autonomous cars, can only be evaluated by expensive numerical or physical experiments, requiring an efficient method with minimum evaluations.

To increase the convergence rate, a sequential Bayesian experimental design framework targeting statistical expectation was developed in [2, 3]. Specifically, they use the Gaussian process regression (GPR) as the surrogate and greedily select the next-best sample one by one which maximizes the information-theoretic acquisition, i.e., the information gain of adding the next sample. Although [2] shows that the proposed method works in several synthetic and practical cases, its sequential nature does bring two drawbacks. Firstly, the samples need to be evaluated one by one, making the duration of the whole process remarkably long. In contrast, the standard Monte Calor determines all samples in the beginning which can maximally utilize parallel computational resources in evaluating samples. Secondly, the determination of samples only focuses on the benefits of the immediate next sample without considering the long-term benefits, for example, the convergence after a certain number of samples.

In this work, we develop a non-myopic batch Bayesian experimental design for statistical expectation. The next batch of samples is selected, which maximizes the long-term information gain (as the acquisition) when they are added together. In addition, we formulate an analytic approximation of the acquisition to facilitate its optimization. The superior performance of the proposed algorithm, in terms of wall time saving and a faster or matched convergence rate than sequential sampling, is demonstrated in a case with arbitrary complex functions generated by RBF kernel and another case using a stochastic oscillator.

## 2   Method

### 2.1   Problem setup

We consider an input-to-response (ItR) system described by a response function $f(\mathbf{x}) : \mathbb{R}^d \to \mathbb{R}$ with $\mathbf{x}$ a $d$-dimensional random input. The input probability $p_{\mathbf{x}}(\mathbf{x})$ is assumed to be known and our objective is the statistical expectation defined as:

$$q = \int f(\mathbf{x}) p_{\mathbf{x}}(\mathbf{x}) \mathrm{d}\mathbf{x}. \tag{1}$$

Submitted to Workshop on Bayesian Decision-making and Uncertainty, 38th Conference on Neural Information Processing Systems (BDU at NeurIPS 2024). Do not distribute.

To compute $q$, we take a Bayesian perspective by placing $f$ a Gaussian process prior $f \sim \mathcal{GP}(0, k(\mathbf{x}, \mathbf{x}'))$ where $k$ is covariance function with hyperparameters $\boldsymbol{\theta}$. Given a dataset $\mathcal{D}_n = \{\mathbf{X}_n, \mathbf{Y}_n\}$ consisting of $n$ inputs $\mathbf{X}_n = \{\mathbf{x}^i \in \mathbb{R}^d\}_{i=1}^n$ and the corresponding outputs $\mathbf{Y}_n = \{f(\mathbf{x}^i) \in \mathbb{R}\}_{i=1}^n$, the underling relation $f$ is predicted as a posterior Gaussian process $f(\mathbf{x})|\mathcal{D}_n \sim \mathcal{GP}(m_n(\mathbf{x}), k_n(\mathbf{x}, \mathbf{x}'))$ with formulae of posterior mean $m_n$ and covariance $k_n$ detailed in Appendix A. The statistical expectation $q|\mathcal{D}_n$ then becomes a random variable with randomness coming from the epistemic uncertainties of $f(\mathbf{x})|\mathcal{D}_n$. Our goal is to choose the most informative batch of samples by optimizing the acquisition function that facilitates convergence of $q$. In the following, we will discuss the form of the acquisition function.

## 2.2 Acquisition function

For selecting the next samples, a popular way is to maximize the information gain (measured by K-L divergence) between the current estimation $q|\mathcal{D}_n$ and hypothetical next estimation $q|\mathcal{D}_n, \tilde{\mathbf{X}}_s, \tilde{\mathbf{Y}}_s$ after adding $s$ number of samples $\tilde{\mathbf{X}}_s$ with responses $\tilde{\mathbf{Y}}_s$ (see [2, 3] for a sequential version):

$$\mathbf{X}_s^* = \operatorname{argmax}_{\tilde{\mathbf{X}}_s} \mathbb{E}\Big[ \int \mathrm{KL}\big(p(q|\mathcal{D}_n, \tilde{\mathbf{X}}_s, \tilde{\mathbf{Y}}_s) \,\|\, p(q|\mathcal{D}_n)\big)\Big],$$
$$\equiv \operatorname{argmax}_{\tilde{\mathbf{X}}_s} \int \mathrm{KL}\big(p(q|\mathcal{D}_n, \tilde{\mathbf{X}}_s, \tilde{\mathbf{Y}}_s) \,\|\, p(q|\mathcal{D}_n)\big) \, p(\tilde{\mathbf{Y}}_s|\tilde{\mathbf{X}}_s, \mathcal{D}_n)\mathrm{d}\tilde{\mathbf{Y}}_s, \tag{2}$$

where $\tilde{\mathbf{Y}}_s$ is chosen based on the current surrogate $f(\mathbf{x})|\mathcal{D}_n$ following a distribution of $\mathcal{N}(\tilde{\mathbf{Y}}_s; m_n(\tilde{\mathbf{X}}_s), k_n(\tilde{\mathbf{X}}_s, \tilde{\mathbf{X}}_s))$. Another more intuitive way is to minimize the variance, as the *predicted* mean squared estimation error (MSE), of $q|\mathcal{D}_n, \tilde{\mathbf{X}}_s, \tilde{\mathbf{Y}}_s$:

$$\mathbf{X}_s^* = \operatorname{argmin}_{\tilde{\mathbf{X}}_s} \mathbb{E}\Big[ \int \operatorname{var}(q|\mathcal{D}_n, \tilde{\mathbf{X}}_s, \tilde{\mathbf{Y}}_s)\Big]$$
$$\equiv \operatorname{argmin}_{\tilde{\mathbf{X}}_s} \int \operatorname{var}(q|\mathcal{D}_n, \tilde{\mathbf{X}}_s, \tilde{\mathbf{Y}}_s) \, p(\tilde{\mathbf{Y}}_s|\tilde{\mathbf{X}}_s, \mathcal{D}_n)\mathrm{d}\tilde{\mathbf{Y}}_s$$
$$\equiv \operatorname{argmin}_{\tilde{\mathbf{X}}_s} \operatorname{var}(q|\mathcal{D}_n, \tilde{\mathbf{X}}_s, \tilde{\mathbf{Y}}_s) \tag{3}$$

where $\operatorname{var}(q|\mathcal{D}_n, \tilde{\mathbf{X}}_s, \tilde{\mathbf{Y}}_s)$ is a constant for $\tilde{\mathbf{Y}}_s$.

Indeed, these two ways are equivalent for estimating the statistical expectation (see detailed derivations in Appendix B), and the optimization finally becomes:

$$\mathbf{X}_s^* = \operatorname{argmax}_{\tilde{\mathbf{X}}_s} \int \mathbf{k}_n(\mathbf{x}, \tilde{\mathbf{X}}_s)p_{\mathbf{x}}(\mathbf{x})\mathrm{d}\mathbf{x}\, \mathbf{K}_n(\tilde{\mathbf{X}}_s, \tilde{\mathbf{X}}_s)^{-1} \int \mathbf{k}_n(\tilde{\mathbf{X}}_s, \mathbf{x})p_{\mathbf{x}}(\mathbf{x})\mathrm{d}\mathbf{x}. \tag{4}$$

While (4) seems straightforward, a numerical integration for the right-hand side can become prohibitively expensive. To make the optimization (likely a high dimensional problem) feasible, we develop an analytical approximation for the acquisition in Appendix C. With the analytical solution, (4) is solved by a multi-start Quasi-Newton method with gradient computed through automatic differentiation in PyTorch[1].

## 2.3 Overall algorithm

We finally show the overall algorithm in Algorithm 1. In each iteration, the number of samples to be selected is specified by $s(t)$ with $t$ the index of iterations. Setting $s(t) = 1$ reduces to the sequential algorithm in [2] and [3]. In this algorithm, one might wonder why we don't schedule all samples initially, considering that the sample responses do not directly appear in (4). Regarding this, we note that the sample responses do influence (4) implicitly via hyperparameters $\boldsymbol{\theta}$ of the Gaussian process. (Should we know the hyperparameters in the beginning, we can determine all samples in one batch where the MSE in (3) is reduced much faster compared with sequential sampling, as shown in Appendix D.) In other words, a sequential algorithm can update the surrogate after each sample, making the selection of the next sample based on a more accurate model (although in a myopic way). The sampling efficiency of the batch algorithm needs to be evaluated in light of the benefits of long-term perspective and the disadvantages of less frequent model updates, which will be demonstrated in the next section.

---

[1]https://github.com/pytorch/pytorch

**Algorithm 1** Batch Bayesian experimental design for statistical expectation

---

**Require:** Number of initial samples $n_{init}$, number of batches $n_{batch}$, number of samples in each batch $s(\cdot)$
**Input:** Initial dataset $\mathcal{D}_{n_{init}} = \{\mathbf{x}^i, f(\mathbf{x}^i)\}_{i=1}^{n_{init}}$
    **Initialization** $t = 0$
    **while** $t < n_{batch}$ **do**
        Train $f(\mathbf{x})|\mathcal{D}_n$
        Solve (4) to find the next-best samples location $\mathbf{X}^*_{s(t)}$
        Implement simulation/experiment to get $f(\mathbf{X}^*_{s(t)})$
        Update the dataset $\mathcal{D}_{n+s(t)} = \mathcal{D}_n \cup \{\mathbf{X}^*_{s(t)}, f(\mathbf{X}^*_{s(t)})\}$
        $t = t + 1, n = n + s(t)$
    **end while**
**Output:** Compute the statistical expectation based on the surrogate.

---

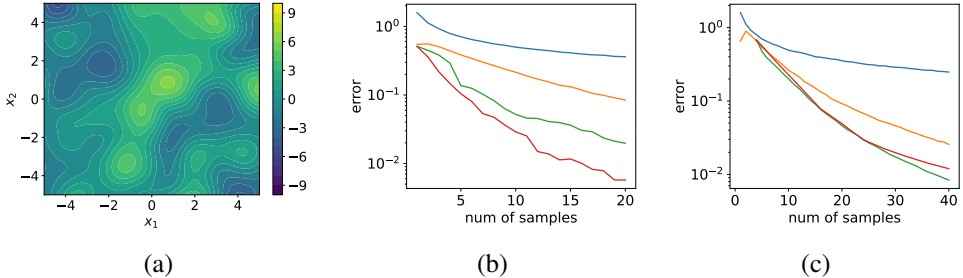

(a)                 (b)                 (c)

Figure 1: (a) an example of two-dimensional RBF functions. Results of RBF functions with (b) known hyperparameters and (c) learned hyperparameters: *random* (——), *random-gpr* (——), *seq-design* (——), and *batch-design* (——) ($s = 4$).

## 3   Results

In this section, we test the performance of the proposed batch design algorithm in two cases: (1) a larger number of complex functions from realizations of Gaussian processes in §3.1, and (2) a stochastic oscillator in §3.2. In each case, we compare the results of batch design (*batch-design*) with sequential design (*seq-design*), direct random sampling (*random*), and random sampling with Gaussian process surrogate (*random-gpr*). For *random*, the expectation is directly computed as the mean of samples, while for *random-gpr* the expectation is computed with a surrogate learned from random samples. The comparison between *random-gpr* and *random* highlights the impact of imposing a prior for $f$, while the advantage of choosing optimal samples over random samples is evidenced in the contrast between *seq-design* and *random-gpr*. Finally, the difference between *batch-design* and *seq-design* measures the effectiveness of picking a group of samples simultaneously instead of a single sample during each iteration.

### 3.1   RBF functions

We firstly test the proposed algorithm in 100 two-dimensional functions constructed from RBF kernel. The hyperparameters for generating these functions are $\boldsymbol{\theta} = \{4, \mathrm{I}_2\}$ (see Appendix A for format of $\boldsymbol{\theta}$) with an example shown in figure 1(a).

The results for a standard Gaussian input $p_{\mathbf{x}}(\mathbf{x})$ with the assumption of known hyperparameters are demonstrated in figure 1(b). Considering there are 100 different functions, we average the error across all functions where, in each function, the error is computed in a root mean squared form of 50 runs considering the randomness in drawing samples. For *seq-design* and *batch-design*, the sampling position is fixed, so we will directly take the fixed error. For *batch-design*, we sample only one batch in the beginning as we assume the hyperparameters are known. From figure 1(b), we can see that methods are ranked in an increasing performance from *random* to *random-gpr* to *seq-design* and finally *batch-design*. That means the prior information is useful and a careful design would also improve the performance. Regarding the design method, the batch design is better than the sequential design as it optimizes all samples as a whole.

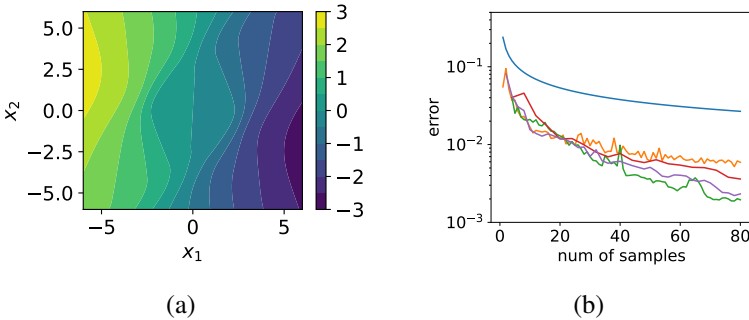

(a)             (b)

Figure 2: (a) response function of the stochastic oscillator. (b) results of *random* (——), *random-gpr* (——), *seq-design* (——), *batch-design* with $s(t) = 4$ (——), *batch-design* with $s(t) = 2$ (——).

We further consider situations where the hyperparameters are unknown with results shown in figure 1(c). For both *batch-design* and *seq-design*, we use 4 initial samples, and the error of each function is also computed in a root mean squared form across different initializations. The *batch-design* with $s(t) = 4$ performs almost the same with *seq-design*, meaning the pro of a non-myopic design is actually offset by the con of fewer hyperparameters updates. But we note that the wall computational time of *batch-design* is only a quarter of *seq-design*.

## 3.2 Stochastic oscillator

We next consider a stochastic oscillator also used in [4, 5, 6]. In particular, the oscillator equation is formulated as

$$\ddot{u}(t) + \delta\dot{u}(t) + F(u) = \xi(t), \tag{5}$$

where $u(t)$ is the state variable, $F$ a nonlinear restoring force. The stochastic process $\xi(t)$, with a correlation function $\sigma_\xi^2 e^{-\tau^2/(2l_\xi^2)}$, is approximated by a two-term Karhunen-Loeve expansion

$$\xi(t) = \sum_{i=1}^{2} x_i \lambda_i \phi_i(t), \tag{6}$$

with $\lambda_i$ and $\phi_i(t)$ respectively the eigenvalue and eigenfunction of the correlation function, $\mathbf{x} \equiv (x_1, x_2)$ is a standard normal variable as the input to the system, satisfying $p_{\mathbf{x}}(\mathbf{x}) = \mathcal{N}(\mathbf{0}, I_2)$ with $I_2$ being a $2 \times 2$ identity matrix. The $F$ term and values of the parameters are kept the same as those in the existing works. The response of the system is considered as the mean value of $u(t; \mathbf{x})$ in the interval $[0, 25]$:

$$f(\mathbf{x}) = \frac{1}{25} \int_0^{25} u(t; \mathbf{x}) \mathrm{d}t, \tag{7}$$

with contour shown in figure 2(a).

We plot the results for different methods in figure 2(b). All results are root mean squared errors with randomness in *random* and *random-gpr* coming from random sampling and randomness in *seq-design* and *batch-design* coming from initializations. For *batch-design*, we test both $s(t) = 2$ and $s(t) = 4$. It demonstrates that *seq-design* performs best among all while *batch-design* with $s(t) = 2$ is almost on par with *seq-design* albeit slightly less efficient.

## 4 Discussion

In this work, we develop a non-myopic batch Bayesian experimental design algorithm for statistical expectation, where the next batch of samples is selected to maximize the information gained (or equivalently to minimize the estimation uncertainty). We apply the results in two test cases, showing that if the hyperparameters (prior) are known, the batch design algorithm converges much faster than the sequential design. For more typical situations requiring learned hyperparameters, the batch design algorithm performs slightly worse, if not equally well, compared to the sequential design. However, it offers substantial savings in wall time. Further tests on additional cases with varying dimensions, complexities, and $s(\cdot)$ are ongoing and will be presented in a full paper.

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

# A   Gaussian process regression

In this section, we briefly introduce the Gaussian process regression (GPR) [7], which is a probabilistic machine learning approach. Consider the task of inferring the underline relatioin $f$ from dataset $\mathcal{D}_n = \{\mathbf{X}_n, \mathbf{Y}_n\}$ consisting of $n$ inputs $\mathbf{X}_n = \{\mathbf{x}^i \in \mathbb{R}^d\}_{i=1}^n$ and the corresponding outputs $\mathbf{Y}_n = \{f(\mathbf{x}^i) \in \mathbb{R}\}_{i=1}^n$. In GPR, a prior, representing our beliefs over all possible functions we expect to observe, is placed on $f$ as a Gaussian process $f(\mathbf{x}) \sim \mathcal{GP}(0, k(\mathbf{x}, \mathbf{x}'))$ with zero mean and covariance function $k$ (usually defined by a radial-basis-function (RBF) kernel):

$$k(\mathbf{x}, \mathbf{x}') = \tau^2 \exp(-\frac{1}{2}((\mathbf{x} - \mathbf{x}')^T \Lambda^{-1} (\mathbf{x} - \mathbf{x}'))), \tag{8}$$

where $\tau$ (characteristic amplitude) and diagonal matrix $\Lambda$ (characteristic length scales) are hyperparameters $\boldsymbol{\theta} = \{\tau, \Lambda\}$ determined by maximizing the likelihood $p(\mathbf{Y}_n)$.

Following the Bayes' theorem, the posterior prediction for $f$ given the dataset $\mathcal{D}$ can be derived to be another Gaussian:

$$f(\mathbf{x})|\mathcal{D} \sim \mathcal{GP}(m_n(\mathbf{x}), k_n(\mathbf{x}, \mathbf{x}')), \tag{9}$$

with mean and covariance respectively:

$$m_n(\mathbf{x}) = \mathbf{k}(\mathbf{x}, \mathbf{X}_n)\mathbf{K}(\mathbf{X}_n, \mathbf{X}_n)^{-1}\mathbf{Y}_n, \tag{10}$$

$$k_n(\mathbf{x}, \mathbf{x}') = k(\mathbf{x}, \mathbf{x}') - \mathbf{k}(\mathbf{x}, \mathbf{X}_n)\mathbf{K}(\mathbf{X}_n, \mathbf{X}_n)^{-1}\mathbf{k}(\mathbf{X}_n, \mathbf{x}'), \tag{11}$$

where matrix element $\mathbf{K}(\mathbf{X}_n, \mathbf{X}_n)_{ij} = k(\mathbf{x}^i, \mathbf{x}^j)$.

# B   Equivalence of (2) and (3)

In this chapter, we will show formulae of (2) and (3) and their equivalence (see a similar conclusion for sequential design in [3]).

We first notice that $q|\mathcal{D}_n$ follows a Gaussian distribution with mean $\mu_1$ and variance $\sigma_1^2$:

$$p(q|\mathcal{D}_n) = \mathcal{N}(q; \mu_1, \sigma_1^2), \tag{12}$$

$$\mu_1 = \mathbb{E}\Big[\int f(\mathbf{x})p_{\mathbf{x}}(\mathbf{x})\mathrm{d}\mathbf{x}|\mathcal{D}_n\Big]$$

$$= \int m_n(\mathbf{x})p_{\mathbf{x}}(\mathbf{x})\mathrm{d}\mathbf{x}, \tag{13}$$

$$\sigma_1^2 = \mathbb{E}\Big[\big(\int f(\mathbf{x})p_{\mathbf{x}}(\mathbf{x})\mathrm{d}\mathbf{x}\big)^2|\mathcal{D}_n\Big] - \mathbb{E}\Big[\big(\int f(\mathbf{x})p_{\mathbf{x}}(\mathbf{x})\mathrm{d}\mathbf{x}\big)|\mathcal{D}_n\Big]^2$$

$$= \iint k_n(\mathbf{x}, \mathbf{x}')p_{\mathbf{x}}(\mathbf{x})p_{\mathbf{x}}(\mathbf{x}')\mathrm{d}\mathbf{x}'\mathrm{d}\mathbf{x}. \tag{14}$$

After adding $s$ hypothetical samples $\{\tilde{\mathbf{X}}_s, \tilde{\mathbf{Y}}_s\}$, $f$ follows an updated distribution $f(\mathbf{x})|\mathcal{D}_n, \tilde{\mathbf{X}}_s, \tilde{\mathbf{Y}}_s \sim \mathcal{GP}(m_{n+s}(\mathbf{x}), k_{n+s}(\mathbf{x}, \mathbf{x}'))$ with

$$m_{n+s}(\mathbf{x}) = m_n(\mathbf{x}) + \mathbf{k}_n(\mathbf{x}, \tilde{\mathbf{X}}_s)\mathbf{K}_n(\tilde{\mathbf{X}}_s, \tilde{\mathbf{X}}_s)^{-1}(\tilde{\mathbf{Y}}_s - m_n(\tilde{\mathbf{X}}_s)), \tag{15}$$

$$k_{n+s}(\mathbf{x}, \mathbf{x}') = k_n(\mathbf{x}, \mathbf{x}') - \mathbf{k}_n(\mathbf{x}, \tilde{\mathbf{X}}_s)\mathbf{K}_n(\tilde{\mathbf{X}}_s, \tilde{\mathbf{X}}_s)^{-1}\mathbf{k}_n(\tilde{\mathbf{X}}_s, \mathbf{x}'). \tag{16}$$

The quantity $q|\mathcal{D}_n, \tilde{\mathbf{X}}_s, \tilde{\mathbf{Y}}_s$ can then be represented by another Gaussian with mean $\mu_2$ and variance $\sigma_2^2$:

$$p(q|\mathcal{D}_n, \tilde{\mathbf{X}}_s, \tilde{\mathbf{Y}}_s) = \mathcal{N}(q; \mu_2(\tilde{\mathbf{X}}_s, \tilde{\mathbf{Y}}_s), \sigma_2^2(\tilde{\mathbf{X}}_s)), \tag{17}$$

$$\mu_2(\tilde{\mathbf{X}}_s, \tilde{\mathbf{Y}}_s) = \int m_{n+s}(\mathbf{x})p_{\mathbf{x}}(\mathbf{x})\mathrm{d}\mathbf{x}$$

$$= \mu_1 + \int \mathbf{k}_n(\mathbf{x}, \tilde{\mathbf{X}}_s)p_{\mathbf{x}}(\mathbf{x})\mathrm{d}\mathbf{x}\, \mathbf{K}_n(\tilde{\mathbf{X}}_s, \tilde{\mathbf{X}}_s)^{-1}(\tilde{\mathbf{Y}}_s - m_n(\tilde{\mathbf{X}}_s)), \tag{18}$$

$$\sigma_2^2(\tilde{\mathbf{X}}_s) = \iint k_{n+s}(\mathbf{x}, \mathbf{x}')p_{\mathbf{x}}(\mathbf{x})p(\mathbf{x}')\mathrm{d}\mathbf{x}'\mathrm{d}\mathbf{x}$$

$$= \sigma_1^2 - \int \mathbf{k}_n(\mathbf{x}, \tilde{\mathbf{X}}_s)p_{\mathbf{x}}(\mathbf{x})\mathrm{d}\mathbf{x}\, \mathbf{K}_n(\tilde{\mathbf{X}}_s, \tilde{\mathbf{X}}_s)^{-1}\int \mathbf{k}_n(\tilde{\mathbf{X}}_s, \mathbf{x})p_{\mathbf{x}}(\mathbf{x})\mathrm{d}\mathbf{x}. \tag{19}$$

With (12) and (17), one can simplify the objective function in (2):

$$\int \mathrm{KL}\big(p(q|\mathcal{D}_n, \tilde{\mathbf{X}}_s, \tilde{\mathbf{Y}}_s) \,\|\, p(q|\mathcal{D}_n)\big)\, p(\tilde{\mathbf{Y}}_s|\tilde{\mathbf{X}}_s, \mathcal{D}_n)\mathrm{d}\tilde{\mathbf{Y}}_s$$

$$= \iint p(q|\mathcal{D}_n, \tilde{\mathbf{X}}_s, \tilde{\mathbf{Y}}_s)\log\frac{p(q|\mathcal{D}_n, \tilde{\mathbf{X}}_s, \tilde{\mathbf{Y}}_s)}{p(q|\mathcal{D}_n)}\mathrm{d}q\, p(\tilde{\mathbf{Y}}_s|\tilde{\mathbf{X}}_s, \mathcal{D}_n))\mathrm{d}\tilde{\mathbf{Y}}_s$$

$$= \int \Big(\log(\frac{\sigma_1}{\sigma_2(\tilde{\mathbf{X}}_s)}) + \frac{\sigma_2^2(\tilde{\mathbf{X}}_s)}{2\sigma_1^2} + \frac{(\mu_2(\tilde{\mathbf{X}}_s, \tilde{\mathbf{Y}}_s) - \mu_1)^2}{2\sigma_1^2} - \frac{1}{2}\Big)p(\tilde{\mathbf{Y}}_s|\tilde{\mathbf{X}}_s, \mathcal{D}_n)\mathrm{d}\tilde{\mathbf{Y}}_s$$

$$= \log(\frac{\sigma_1}{\sigma_2(\tilde{\mathbf{X}}_s)}) + \frac{1}{2\sigma_1^2}\big(\sigma_2^2(\tilde{\mathbf{X}}_s) - \sigma_1^2 + \int (\mu_2(\tilde{\mathbf{X}}_s, \tilde{\mathbf{Y}}_s) - \mu_1)^2 p(\tilde{\mathbf{Y}}_s|\tilde{\mathbf{X}}_s, \mathcal{D}_n)\mathrm{d}\tilde{\mathbf{Y}}_s\big)$$

$$= \log(\frac{\sigma_1}{\sigma_2(\tilde{\mathbf{X}}_s)}) + \frac{1}{2\sigma_1^2}\big(\sigma_2^2(\tilde{\mathbf{X}}_s) - \sigma_1^2$$

$$\qquad + \int \mathbf{k}_n(\mathbf{x}, \tilde{\mathbf{X}}_s)p_{\mathbf{x}}(\mathbf{x})\mathrm{d}\mathbf{x}\, \mathbf{K}_n(\tilde{\mathbf{X}}_s, \tilde{\mathbf{X}}_s)^{-1}\int \mathbf{k}_n(\tilde{\mathbf{X}}_s, \mathbf{x})p_{\mathbf{x}}(\mathbf{x})\mathrm{d}\mathbf{x}\big)$$

$$= \log(\frac{\sigma_1}{\sigma_2(\tilde{\mathbf{X}}_s)}). \tag{20}$$

Since $\sigma_1$ does not depend on $\tilde{\mathbf{X}}_s$, (2) can be reformulated as

$$\mathbf{X}_s^* = \mathrm{argmin}_{\tilde{\mathbf{X}}_s}\, \sigma_2^2(\tilde{\mathbf{X}}_s) \tag{21}$$

$$= \mathrm{argmax}_{\tilde{\mathbf{X}}_s}\, \int \mathbf{k}_n(\mathbf{x}, \tilde{\mathbf{X}}_s)p_{\mathbf{x}}(\mathbf{x})\mathrm{d}\mathbf{x}\, \mathbf{K}_n(\tilde{\mathbf{X}}_s, \tilde{\mathbf{X}}_s)^{-1}\int \mathbf{k}_n(\tilde{\mathbf{X}}_s, \mathbf{x})p_{\mathbf{x}}(\mathbf{x})\mathrm{d}\mathbf{x}, \tag{22}$$

where (21) is exactly (3). The final optimization problem (22) ((4) in §2.2) is obtained by substituting (19) into (21).

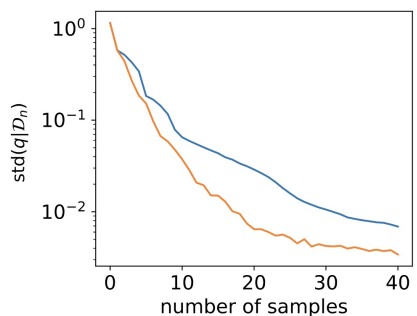

Figure 3: The standard deviation of $q|\mathcal{D}_n$ computed by sequential design (——) and batch design (——) for Gaussian input $\mathbf{x} \sim \mathcal{N}(\mathbf{0}, \mathrm{I}_2)$ and known hyperparameters $\boldsymbol{\theta} = \{4, \mathrm{I}_2\}$.

## C    Analytical approximation of (4)

In computing the right-hand side of (4), the heaviest computation involved is the integral $\int k_n(\tilde{\mathbf{x}}, \mathbf{x})p_{\mathbf{x}}(\mathbf{x})\mathrm{d}\mathbf{x}$. Expanding $k_n$ with (11), we have:

$$\int k_n(\tilde{\mathbf{x}}, \mathbf{x})p_{\mathbf{x}}(\mathbf{x})\mathrm{d}\mathbf{x} = \mathcal{K}(\tilde{\mathbf{x}}) - \mathbf{k}(\tilde{\mathbf{x}}, \mathbf{X}_n)\mathbf{K}(\mathbf{X}_n, \mathbf{X}_n)^{-1}\mathcal{K}(\mathbf{X}_n), \tag{23}$$

with

$$\mathcal{K}(\mathbf{x}) = \int k(\mathbf{x}, \mathbf{x}')p_{\mathbf{x}}(\mathbf{x}')\mathrm{d}\mathbf{x}', \tag{24}$$

If the input $\mathbf{x}$ is Gaussian with mean $\mathbf{w}$ and covariance $\Sigma$, (24) has analytical expression for RBF kernel with characteristic length $\Lambda$:

$$\int k(\mathbf{x}, \mathbf{x}')\mathcal{N}(\mathbf{x}; \mathbf{w}, \Sigma)\mathrm{d}\mathbf{x}' = |\Sigma\Lambda^{-1} + \mathrm{I}|^{-\frac{1}{2}}k(\mathbf{x}, \mathbf{w}; \Sigma + \Lambda). \tag{25}$$

To make $\mathcal{K}$ analytically tractble for arbitrary $p_{\mathbf{x}}(\mathbf{x})$, we approximate $p_{\mathbf{x}}(\mathbf{x})$ with the Gaussian mixture model [8] with $n_{GMM}$ Gaussian functions:

$$p_{\mathbf{x}}(\mathbf{x}) \approx \sum_{i=1}^{n_{GMM}} \alpha_i\mathcal{N}(\mathbf{x}; \mathbf{w}_i, \Sigma_i). \tag{26}$$

(24) can then be formulated as:

$$\mathcal{K}(\mathbf{x}) \approx \sum_{i=1}^{n_{GMM}} \alpha_i\int k(\mathbf{x}, \mathbf{x}')\mathcal{N}(\mathbf{x}'; \mathbf{w}_i, \Sigma_i)\mathrm{d}\mathbf{x}'$$
$$= \sum_{i=1}^{n_{GMM}} \alpha_i|\Sigma_i\Lambda^{-1} + \mathrm{I}|^{-\frac{1}{2}}k(\mathbf{x}, \mathbf{w}_i; \Sigma_i + \Lambda). \tag{27}$$

## D    Potential improvement on sampling efficiency

In this section, we show the accelerated reduction of MSE in (3) for batch sampling over sequential sampling. This serves as theoretical evidence of the potential for enhanced sampling efficiency.

In figure 3, we plot the standard deviation of $q|\mathcal{D}_n$ (square root of MSE in (3)) with batch design (one batch) and sequential design for standard Gaussian input and known hyperparameters $\boldsymbol{\theta} = \{4, \mathrm{I}_2\}$ with $\mathrm{I}_2$ being a $2 \times 2$ identity matrix. It shows the batch design performs much better than the sequential design which is anticipated as we get the 'free lunch'—the benefits of a long-term perspective without any side effects from fewer model updates. The sampling positions of sequential design and batch design are plotted in figure 4. The batch samples show beautiful symmetric structures fitting the symmetric input and hyperparameters. In contrast, sequential samples show a strong greedy pattern. For example, when we have three samples, the sequential samples clearly favor one direction while batch samples form an equilateral triangle. (**see next page for figure 4**)

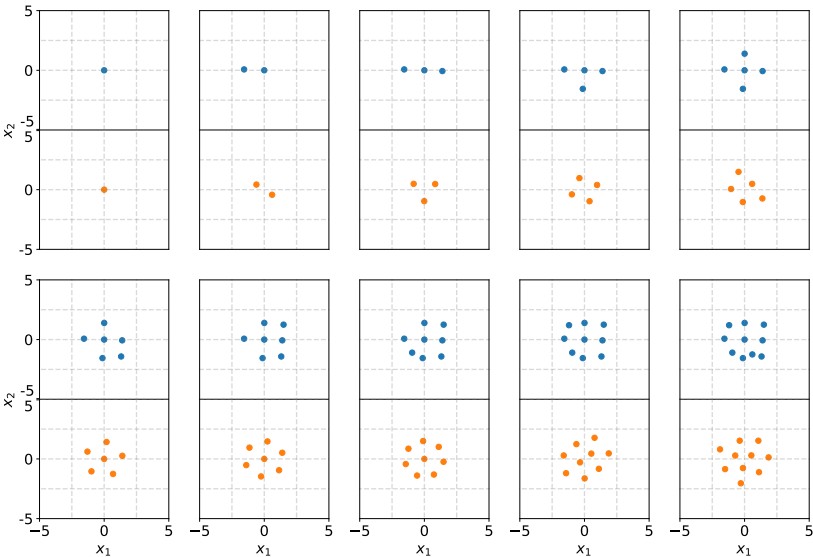

Figure 4: Sampling position of sequential design (●) and batch design (●) for Gaussian input $\mathbf{x} \sim \mathcal{N}(\mathbf{0}, \mathrm{I}_2)$ and known hyperparameters $\boldsymbol{\theta} = \{4, \mathrm{I}_2\}$.

