# OpenReview forum: "Non-Myopic Batch Bayesian Experimental Design for Quantifying Statistical Expectation"
_NeurIPS.cc/2024/Workshop/BDU — Submitted to NeurIPS BDU Workshop 2024_

### Official Review · Reviewer_Ytpi · 2024-09-25
**Non-Myopic Batch Bayesian Experimental Design for Quantifying Statistical Expectation**

**Rating:** 5
**Confidence:** 4

**Review:**

## Summary
The authors propose an acquisition function that ideally chooses a batch of locations to evaluate the underlying process at, such that, in expectation, the resulting surrogate converges to the same surrogate that would be learned in a purely single observation sequential fashion. The acquisition function they propose seeks a batch of new points that maximizes the information gain, provided that the current model is correctly specified. This proposed strategy seeks to increase the rate of convergence of the surrogate model to the ground truth.

## Review
The introduction is ok and briefly describes what Bayesian optimization seeks to achieve, without explicitly referencing it--which is ok. I think the submission would've benefitted from an abstract. The arguments presented for their acquisition function felt rushed and the writing doesn't demonstrate any clear intuition.

I think the authors could've motivated why the batch design performs better than the sequential design. They note "baatch samples show beautiful symmetric structures fitting the symmetric input and hyperparameters"--should we care about this? Is this particularly important or just a nice observation?

The "strong greedy pattern" they mentioned wasn't evidently clear to me based solely on the plots. The patterns seemed to deviate minimally, especially as the batch size increased. Also, since the sequential strategy plots weren't symmetric, why does this imply greediness? Or vice-versa? The graphs seem to show similar collective behavior as the samples increase, so the point in this section was lost on me.

In algorithm 1, what do we do with $\mathcal{D}_{n + s(t)}$? We construct this dataset for each distribution and have a posterior model for each batch of observations, I assume. Do we then compute the statistical expectation based on the average of all these "fantasized" surrogates or do we choose the best one?

I did enjoy the experiments that were conducting on a reasonably large number of 2D functions. It would be nice to see it's effects on a few other problems in, maybe, 1D-6D, just to build some intuition behind how this method performs in higher dimensional spaces. Maybe there is something that can be inferred by looking at a common metric across various dimensionalities.

The algorithm also appears to be missing some details. I could've interpreted it incorrectly. It would benefit from coupling "compute the statistical..." in the output of the algorithm back to the optimization problem the authors care about. The algorithm appeared to be solving a sub-problem required to even consider the primary problem. Though I'm not sure my interpretation here is fully correct.

---

### Official Review · Reviewer_jNoN · 2024-09-26
**Non-Myopic Batch Bayesian Experimental Design for Quantifying Statistical Expectation**

**Rating:** 4
**Confidence:** 3

**Review:**

The paper is well-written and easy to follow. However, the batch setting for Bayesian experimental design has been discussed in various papers (some listed below and others exist), and none of the existing literature is mentioned in this paper. I would recommend that the authors carefully characterise the differences between their proposed methodology and the existing literature, both theoretically and experimentally. In its current form, I do not believe the proposed methodology is novel. Furthermore, the term "non-myopic" usually refers to algorithms that plan multiple steps into the future for experimental design and Bayesian optimization. This doesn't seem to be the case for this algorithm, where points are selected greedily after each step.


"Batch simulations and uncertainty quantification in Gaussian process surrogate approximate Bayesian computation"
"Bayesian sequential optimal experimental design for nonlinear models using policy gradient reinforcement learning"

---

### Decision · Program_Chairs · 2024-10-09

**Decision:**

Reject

**Comment:**

Reviews for this paper are mixed, with complains including insufficient attention given to previous work on this topic. My general take from reading the reviews is that the paper is not at a sufficiently-finished state to warrant presentation at a workshop. I encourage the authors to continue their work and resubmit to a future workshop.